# How does it all end? Trends and disparities in health at the end of life

Yana C. Vierboom *

Max Planck Institute for Demographic Research, Rostock, Germany

* vierboom@demogr.mpg.de

## Abstract

### Objectives

To consider trends and disparities in end-of-life health in the US.

### Methods

I use data from the National Health Interview Survey, linked to death records through 2015, for respondents who died at ages 65+ to compare the prevalence of three health outcomes in the last six years of life across time, sex, age, race, and educational attainment. Self-rated health (SRH) is available for respondents interviewed in years 1987–2014, while information on activities of daily living (ADL) and instrumental activities of daily living (IADL) is available for the period 1997–2014.

### Results

By the end of the study period, individuals reported two fewer months of fair/poor health at the end of life than those dying in earlier years. In contrast, time lived with at least one activity limitation at the end of life generally remained comparable. Compared to men, women on average reported an additional year of living with an IADL limitation before death, and an additional eight months with an ADL limitation. Despite sex differences in disability, both sexes reported similar periods of fair/poor SRH before death. Similarly, while individuals who lived to older ages experienced a longer disabled period before death than individuals who died at younger ages, all age groups were equally likely to report fair/poor SRH. Black adults and adults with less formal schooling also spent more time with an end-of-life disability. For men, these racial and socioeconomic disparities lessened as death approached. For women, inequalities persisted until death.

### Discussion

These findings suggest that despite increasing life expectancy, the period of poor health and disability prior to death has not recently been extended. Black women and women with less than a high school degree, require extended support at the end of life.

**Data Availability Statement:** All data is publicly-available and, along with all Stata code for analysis, is available on the author's webiste: https://yanavierboom.weebly.com/replication-materials.html.

**Funding:** The author received no specific funding for this work.

**Competing interests:** The author has declared that no competing interests exist.

## Introduction

The period preceding death has become an important and distinct stage of the contemporary life course [1,2]. Where death was once sudden, the sweeping health innovations of the last 150 years mean that death in the United States today often follows an extended period of chronic illness [1]. Although each death is influenced by a unique combination of social, behavioral, and genetic factors, there are also commonalities across this final stage. For many adults, disability and depression increase at the end of life, while cognitive skills and a sense of well-being begin to wane [3–6]. Typically, self-rated health, vision, handgrip strength, and weight also decline toward the end of life [4,5,7,8].

Rising life expectancy at older ages has raised concerns that the period of poor health and disability prior to death is growing. Research typically addresses this topic with the implicit assumption that advancing age is the main risk factor for declining health. However, the onset of several health conditions, including end-of-life depression and cognitive decline, is more closely linked to years of life remaining than years lived [4,7,8]. Comparing the health of older adults who are the same proximity to death (for example, comparing all adults in their last year of life) may yield different insights than comparing adults who are the same age, but differing distances from death (for example, comparing all 70 year-olds).

In this paper, I examine trends and inequalities in aging from the perspective of time to death, rather than time since birth. I compare three indictors of health—self-rated health (SRH) and two self-reports of disability—in the last 6 years of life among adults dying at ages 65+ across time, sex, age, race, and educational attainment. SRH is a subjective and self-reported indicator of health. While the two disability measures are also self-reported, they serve as more objective assesments of requiring assistance. This study is the first to examine annual trends in SRH at the end of life, as well as the first to produce national estimates of end-of-life SRH for several subpopulations. Quantifying end-of-life processes is crucial to both the success of programs aiming to meet the needs of a growing older population and to empower individuals to create advanced care plans about the end-of-life care they wish to receive.

## Background

### Years-to-death

From evaluating the financial wellbeing of pension systems to predicting a population's health-care needs, the end-of-life period is of interest across disciplines. A significant analytic decision in this research is how to measure time. While most research considers the time elapsed since birth, some approaches measure backward from the other end of the lifespan: death. Years to death can be a proxy for the complex and interacting social, behavioral, environmental, and genetic processes that determine each individual's moment of death. The usefulness of a variable for remaining lifetime was first described in the 1970's (see Sanderson & Scherbov [9] for a history of the variable, as well as a demonstration of using the variable to study population aging). An allure of the variable is that its utility remains under-explored, despite yielding new perspectives that can be missed if using only chronological age.

An important analytic decision when using the variable is the maximum length of the retrospective period before death. While some of the studies cited throughout this paper consider the last one or two years of life [10–12], others extend 3–8 years before death [6,13–16], and some well beyond 10 years [4,7,8]. Lunney et al.'s [15] finding that racial disparities in disability are "erased" in the last 1–1.5 years of life suggests that a period longer than 2 years before death is needed to capture evolving patterns of disparities.

Gerstorf et al. [4] find that well-being among older adults in several countries begins to decline 3–5 years before death, around the same time as cognitive abilities [6]. Stenholm et al. [8] find that, compared to similarly-aged respondents who did not die, deceased participants of the Health and Retirement Study had a higher prevalence of poor SRH as early as 11–12 years before death. In a small sample of males ages 60+, Alley et al. [7] document that weight loss typically begins as early as nine years prior to death. Raab et al. [16] examine tandem trajectories of mental health and disability in the last eight years of life, while Gill et al. [10] document five disability trajectories in the last year of life. Although most individuals in Gill et. al's sample were not disabled 12 months before death, more than half were severely disabled in the last month.

## Trends in healthy aging

As individuals age, many develop at least one chronic condition. One approach for estimating the impact of morbidity on day-to-day functioning is to determine whether an individual has difficulty performing Activities of Daily Living (ADLs) or Instrumental Activities of Daily Living (IADLs). ADL's include basic tasks such as dressing and eating, while IADL's encompass activities that facilitate independent living, like grocery shopping or balancing a checkbook. ADL limitations are strong indicators of requiring physical assistance, with roughly 40% of community-dwelling adults age 65+ with one limitation and nearly 90% with 3+ receiving caregiving help [17]. While IADL limitations are less disabling than ADL limitations, an IADL limitation indicates that an individual requires some level of support in order to live independently.

An influential extraneous force shaping trends in disability prevalence is the changing composition of the population. Given the sweeping changes of the twentieth century, younger cohorts are reaching older ages having had better childhood health and more educated parents, reduced exposure to physically-demanding jobs, and higher levels of educational attainment—all factors linked to postponed age at onset of limitations [18]. More recent cohorts of older men are also less likely than their predecessors to be heavy smokers [19]. Because cohorts are evolving at the same time as the contexts in which they live, it is difficult to separate period and cohort effects. Crimmins et al. [20] speculate that once-disabling conditions may be less disabling today due to factors such as earlier diagnosis and better disease management, improved housing environments, and technological changes.

The age-specific prevalence of some disabilities may be declining over time, though findings are sensitive to analytic choices. Although the age-specific prevalence of ADL limitations declined in the 1990's [18,21,22], evidence on whether more recent cohorts are less likely to experience IADL limitations is conflicting [18,22]. Other work using broader measures of disability suggests that recent increases in life expectancy at age 65 were primarily driven by increases in disability-free years [23].

Research on trends in health and disability by years of life remaining, rather than years lived, is limited. In a working paper, Cutler et al. [14] find that the prevalence of limitations in the last 5 years of life declined by up to 14% in the early 1990's, but that the trend remained flat the following decade. This latter finding is echoed by Smith et al. [12], who find no trend in the prevalence of disability in the last two years of life for decedents dying between 1995–2010. In contrast, Beltrán-Sánchez et al. [13] find that the cohort of people dying in the late 2000's reported a higher prevalence of chronic conditions in the final six years of life than did the cohort dying between 1998–2004. The increase in chronic conditions at the end of life may be a recent phenomenon, as Cutler et al. [14], using slightly older data, find no significant change throughout the 1990's in the prevalence of major chronic conditions in the last three years of life.

## Inequalities in healthy aging

The processes translating health inequalities at younger ages into inequalities at older ages are nuanced. On one hand, inequalities in health may be magnified with age. Adverse health experiences might accumulate over the life course and interact with vulnerabilities that accompany old age. The implications of socially-patterned health behaviors from younger years could also be postponed to older ages, such as the lag between cigarette smoking and the onset of lung cancer. On the other hand, not even the most privileged groups are exempt from aging—a fact which may level health inequalities as age advances [24,25]. Another possibility through which inequalities may diminish with age is selective mortality. Since some populations are exposed to systematically higher mortality rates throughout their lives, these groups can be highly select by the time they reach older ages. By nature of their design, studies that use chronological age (comparing 80 year-old White adults to their 80 year-old Black peers, for example) ignore the influence of selective mortality. While the issue is greatly lessened when considering time-to-death (comparing racial differences five years before death, for example), it nevertheless persists anytime a study sample has a minimum age below which differential mortality occurs.

Generally, conclusions about how health inequalities evolve across the life course depend on whether time is measured by elapsed age or proximity to death. Although older Black adults experience a higher prevalence of disability compared to their White peers of the same age [15,26,27], Lunney et al. [15] find that Black-White differences in disability are "erased in the final 1 to 1.6 years before death", suggesting that "dying eliminate[s] a health disparity." This result is consistent with work finding no racial differences in disability in the two years before death [12]. Some racial disparities, however, persist. Raab et al. [16] find that Black decedents are more likely than their White peers to exhibit adverse combinations of disability and poor mental health in the last eight years of life. Warner and Brown [27] test for several possible explanations, including the mediating roles of adult socio-economic status, health behaviors, and marital status. The authors find that while these variables account for Black-White differences in limitations for men, they do not fully explain the disadvantage for Black women. In a study similar to the present one, Liao et al. [11] find while Black-White differences in outcomes like long-term disability and hospital stays are persistent over the last two years of life, they are mostly explained by differences in educational attainment. In a review of research on end-of-life quality, Carr & Luth [1] examine how significant racial and socio-economic disparities in creating end-of-life plans may influence disparities in quality of life before death.

Just as certain age-specific racial disparities in health, disparities by educational attainment in health remain at older ages. Educational differences in the number of healthy years an average individual could expect to live in the 1980's and 1990's were even larger than differences in overall life expectancy [28] and Gini coefficients for health inequalities by education have been shown to increase with age [29]. Educational disparities in the age of onset of ADL limitations have also widened since the 1990's [30]. Unlike the leveling effect impending death has on some racial disparities in disability by time-to-death, socio-economic differences appear to persist into the final stage of life. High school dropouts are more likely to be disabled in the last two years of life than high school graduates [12], and individuals with a terminal high school degree or less are more likely than those with at least some college education to report a combination of disability and poor mental health at the end of life [16].

## Methods

### Data

I use data from the 1987–2014 National Health Interview Survey (NHIS), downloaded from IPUMS [31]. The NHIS, conducted annually by the National Center for Health Statistics

(NCHS), is a cross-sectional health survey of the civilian, non-institutionalized U.S. population that has been linked to death records through the end of 2015. While residents of long-term care institutions are not included in the baseline sample, the sample may include individuals who were interviewed at home and then moved to an institution during the follow-up period. More information on the survey is available on the IPUMS website (nhis.ipums.org).

The NHIS began consistently asking respondents about ADL and IADL limitations in 1997, but has included an annual question on SRH since the 1970's. Since respondent matching to death records did not begin until 1986, however, and because a different weighting scheme was used in 1986, I begin the analysis of SRH trends in 1987. The analysis of ADL and IADL measures begins in 1997. I examine all outcomes until 2014, the last year for which interviewed respondents in the public-use files have been linked to death records (through 2015).

## The sample

I restrict the sample to respondents who died at age 65 or above, within 6 years of being interviewed. I estimate the prevalence of each health outcome for all respondents not missing information on the outcome variable. The SRH analysis sample consists of 77,295 individuals across all years 1987–2014. Sample sizes for the IADL and ADL analyses, which span years 1997–2014, are 40,354 and 40,359, respectively.

## Years-to-death

I assign each decedent a value for years to death by subtracting the calendar year in which a respondent was interviewed from the respondent's year of death reported in the linked mortality file.

I consider the last six years of life. Since the annual trends portion of the analysis requires six years to have elapsed since interview and mortality data is only available through 2015, the last year for which estimates can be produced is for respondents interviewed in 2008. Although a window longer than six years would be optimal, it strikes the balance between observing outcomes and disparities for as long as possible while tracking relatively recent trends. The categorical variable for years to death ranges from 1 (decedents died within 0–11.9 months of being interviewed) to 6 (decedents died within 5–5.9 years of interview).

## Health outcomes

1. Self-rated health is a predictor of subsequent mortality (see Jylhä [32] for why this may be). The NHIS asks respondents to categorize their health as excellent, very good, good, fair, or poor. For some of the analyses, I dichotomize answers into a dummy variable for unfavorable health (fair or poor SHR).

2. The NHIS asks questions on six different ADLs: whether a respondent requires help eating, bathing, dressing, moving about the home, using the toilet, or getting in/out of bed. Consistent with existing research, I classify an individual as having a disability if s/he requires help in performing at least one of these activities [12,22]. In the Supplemental Materials, I provide more fine-grained results for individuals with 1, 2, or 3+ ADLs.

3. Though IADL limitations are less disabling than ADL limitations, their presence indicates that an individual requires some support in order to live independently. The NHIS ascertained information on IADL limitations in the NHIS using a single yes/no question for whether a respondent needed help for "handling routine needs, such as everyday household chores, doing necessary business, shopping, or getting around for other purposes." While

the above ADL variable combines six survey questions, the IADL variable reflects only this single question.

## Population characteristics

I consider patterns in end-of-life health across three socially-stratifying characteristics that are linked to differential health outcomes across the life course: sex, race (non-Hispanic Black, non-Hispanic White), and educational attainment (<high school, high school degree or some college/associate's degree, bachelor's degree or more). I also compare the outcomes of those dying at different ages (65–74, 75–84, and 85+). Due to the small sample sizes for other racial/ethnic categorizations, I limit the racially-stratified analysis to Black-White comparisons. However, the other comparisons include all respondents, regardless of reported race/ethnicity.

## Analytic approach

I estimate the proportion of the population reporting a given health outcome at each year $x$ before death. This proportion can be interpreted as the proportion of total person-years lived with the given health outcome in year $x$ before death—or as the average proportion of year $x$ each individual spends in the health state. A 0.4 annual prevalence of poor SRH, for example, indicates that 40% of the population reported poor health, that 40% of all person-years lived were in poor health, and that, on average, each individual spent 40% of the year, or approximately 5 months, in poor health. To estimate the total number of the final six years that are spent in poor health, I conceptualize respondents as belonging to a synthetic cohort assembled along years to death and sum the prevalence across each year to death.

To compare outcomes across population subgroups, I estimate the mean of an outcome prevalence across the study period (years 1997–2014 for SRH; years 1997–2014 for disability).

For the time trends analysis, I stratify results across two-year interview periods. Since respondents interviewed in later years were not exposed for the full six years, I limit the sample to respondents interviewed before 2009 for this portion of the analysis. To estimate 95% confidence intervals for the sums of averages, I use bootstrapping procedures to repeat the estimation 500 times and take the 2.5th and 97.5th centiles.

All analyses were performed using Stata version 16 (Statacorp), account for the complex survey design of the NHIS using the *svy* package, and employ the NCHS-recommended weights (*mortwt* in IPUMS). Data and code for replication are available on the author's webpage.

## Results

Table 1 shows the nationally-representative distribution of population characteristics and the mean time between interview and death for each subgroup. The table also reports the median ages at death (median instead of mean, as age is top-coded). Of those who survive to age 65, most adults also survive to age 80, with women outliving men by three years (median ages at death of 83.2 vs. 80.3). Among both sexes, non-Hispanic Black individuals tend to die three years earlier than their White peers. Notably, women with less than a high school education in this population die at older ages than college-educated women (median ages of death of 84.1 vs. 82.7). Since levels of educational attainment among U.S. women increased considerably over time, women who are oldest at interview—and therefore older at death—are disproportionately less educated, which results in their paradoxically older median age at death (the reverse is true at younger ages). The age at death has climbed since the beginning of the survey,

**Table 1. Characteristics of adults 65+ dying within 6 years of NHIS interview, 1986–2014[a].** Standard deviation in parentheses.

| Characteristic | Females (N = 20,639)[b] | | | Males (N = 19,745)[b] | | |
|---|---|---|---|---|---|---|
| | % | Median age at death[c] | Mean yrs to death[d] | % | Median age at death[c] | Mean yrs to death[d] |
| **Overall** | -- | 83.17 | 3.53 (0.01) | -- | 80.25 | 3.42 (0.01) |
| **Age at death** | | | | | | |
| 65–74 | 22.04 (0.36) | 70.75 | 3.43 (0.03) | 30.29 (0.37) | 70.25 | 3.34 (0.02) |
| 75–84 | 35.93 (0.42) | 80.58 | 3.47 (0.02) | 39.67 (0.37) | 80.25 | 3.43 (0.02) |
| 85+ | 42.03 (0.47) | 85.00 | 3.63 (0.02) | 30.04 (0.40) | 85.00 | 3.49 (0.02) |
| **Race[e]** | | | | | | |
| Non-Hisp. White | 89.24 (0.32) | 83.67 | 3.54 (0.01) | 90.87 (0.28) | 80.67 | 3.41 (0.02) |
| Non-Hisp. Black | 10.76 (0.32) | 80.33 | 3.46 (0.03) | 9.13 (0.28) | 77.08 | 3.46 (0.04) |
| **Educational attainment** | | | | | | |
| <High school (HS) | 33.10 (0.43) | 84.08 | 3.51 (0.02) | 30.94 (0.40) | 80.67 | 3.39 (0.02) |
| HS/Some college | 56.77 (0.42) | 82.50 | 3.57 (0.02) | 50.76 (0.40) | 79.67 | 3.43 (0.02) |
| BA or more | 10.13 (0.28) | 82.67 | 3.48 (0.04) | 18.30 (0.37) | 80.83 | 3.45 (0.03) |
| **Year of interview** | | | | | | |
| 1997–98 | 12.97 (0.28) | 82.25 | 3.69 (0.03) | 12.40 (0.26) | 79.50 | 3.62 (0.04) |
| 1999–00 | 12.32 (0.26) | 82.17 | 3.71 (0.03) | 12.59 (0.27) | 79.50 | 3.59 (0.03) |
| 2001–02 | 12.08 (0.28) | 82.83 | 3.69 (0.03) | 12.79 (0.26) | 80.17 | 3.53 (0.03) |
| 2003–04 | 12.63 (0.28) | 83.42 | 3.67 (0.04) | 12.31 (0.29) | 80.75 | 3.57 (0.04) |
| 2005–06 | 12.61 (0.27) | 83.67 | 3.78 (0.04) | 11.99 (0.27) | 80.58 | 3.67 (0.04) |
| 2007–08 | 12.20 (0.34) | 83.67 | 3.67 (0.04) | 11.99 (0.34) | 80.42 | 3.57 (0.04) |
| 2009–10 | 12.48 (0.34) | 83.58 | 3.62 (0.04) | 12.69 (0.31) | 80.42 | 3.49 (0.04) |
| 2011–12 | 8.55 (0.26) | 83.75 | 2.73 (0.03) | 8.66 (0.22) | 80.00 | 2.78 (0.03) |
| 2013–14 | 4.16 (0.16) | 83.83 | 1.73 (0.03) | 4.58 (0.18) | 80.83 | 1.68 (0.03) |

a. All percentages and proportions weighted using IPUMS mortality weights *mortwt*.

b. Sample sizes in the header reflect the number of respondents in the surveys who died within 6 years of being interviewed, at age 65 or above. The SRH analysis begins in 1987, but for brevity, details are not included in this table. Each set of analyses is additionally restricted to respondents not missing information on the given health outcome. 20,582 females and 19,694 males were not missing information on self-rated health. 20,622 females and 19,732 males were not missing information on ADL limitations. 20,630 females and 19,729 males were not missing information on IADL limitations.

c. Median, rather than mean, age reported since age in the NHIS is top-coded at 85. For this reason, the median age at death for 85+ year-olds is 85.0.

d. The NHIS/IPUMS public-use files include respondents' quarter of death and month of interview. By assuming that respondents were interviewed on the 15th of the month and died half-way though the quarter, I add some precision to the years-to-death variable.

e. Due to small sample sizes, non-White and non-Black respondents are excluded from race-specific analyses (but included in all other estimates).

with those interviewed in 2013–2014 dying about 1.5 years older than those interviewed 15 years earlier.

On average, individuals died about 3.5 years after being interviewed. The mean time to death is shorter for decedents interviewed between 2010 and 2014 simply due to the design of this analysis (the post-interview exposure period is less than six years).

Fig 1 shows time trends by sex in the average time out of the last 6 years of life spent in fair/poor health or with limitations. The accompanying S1 Appendix provides more detailed estimates, with numeric results for each level of SRH and number of limitations. Men and women were equally likely to report adverse health, with both sexes on average spending 2.5–2.8 years of the last 6 in fair/poor health. Though levels fluctuate across interview years, the general trend is toward a slight decline in time spent in adverse health since the 1980's. Compared to participants interviewed in 1987–88, respondents interviewed in 2007–08 spent roughly 2 months less in fair/poor health.

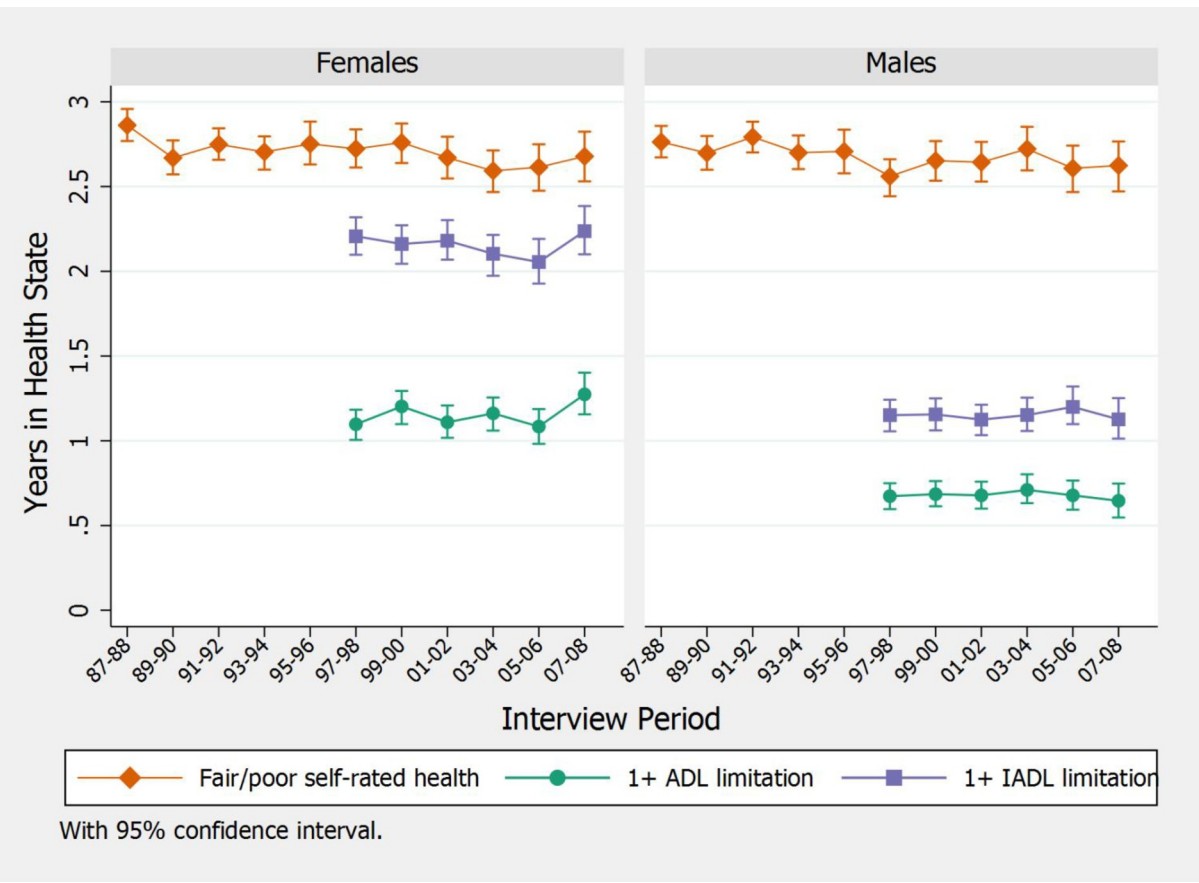

Source: Self-rated health from NHIS 1987-2014, IADL and ADL from NHIS 1997-2014.
95% confidence intervals from 500 bootstrapped replications.

**Fig 1. Years out of last six years of life spent in given health state, over time (with 95% confidence intervals).**

Fig 1 also shows results for time spent with at least one IADL or ADL limitation for decedents interviewed between 1997–2008. Although men and women were equally likely to report adverse health at the end of life, women reported either kinds of limitations for much longer. Men spent just over 1 year requiring help with at least one IADL task, and roughly 8 months with at least one ADL limitation (compared to over 2 years for IADL and 13 months for ADL help among women). While there was no change for men in these measures over the study period, trends are noisier for women. Time spent with either type of limitation increased by two months between the most recent interview periods, disrupting the slight trend toward declining IADL limitations in previous years. S1 Appendix, which divides ADL limitations into 1, 2, and 3+ limitations, suggests that fluctuations in women's time spent with at least 1 ADL limitation were driven by changes in time spent with 2+ limitations.

Figs 2–4 pool data across interview years 1997–2014 and plot the mean proportion of each year spent in each health state, by sex and population characteristic. Dashed lines connect the values for each prevalence to reflect the trajectories experienced by synthetic cohorts. S2 Appendix complements these figures with numeric values and more detailed health outcomes, as well as with results for men and women overall.

The first of these figures, Fig 2, compares decedents who die at ages 65–74, 75–84, and 85+. Regardless of age at death, the prevalence of poor health and disability increase in the years

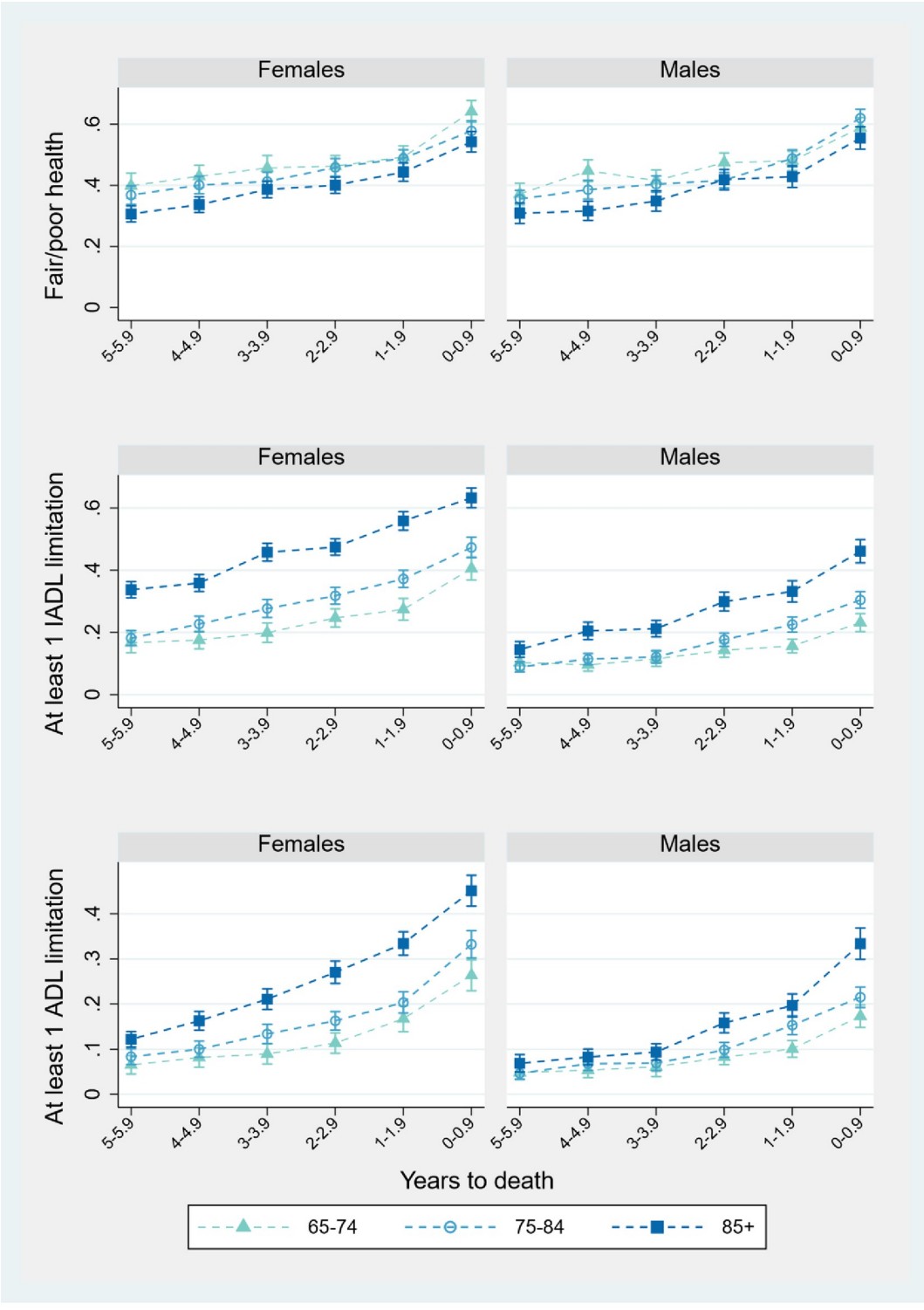

*Source*: Self-rated health from NHIS 1987-2014, IADL and ADL from NHIS 1997-2014

**Fig 2. Proportion of population in given health state across final years of life, by age at death (with 95% confidence intervals).**

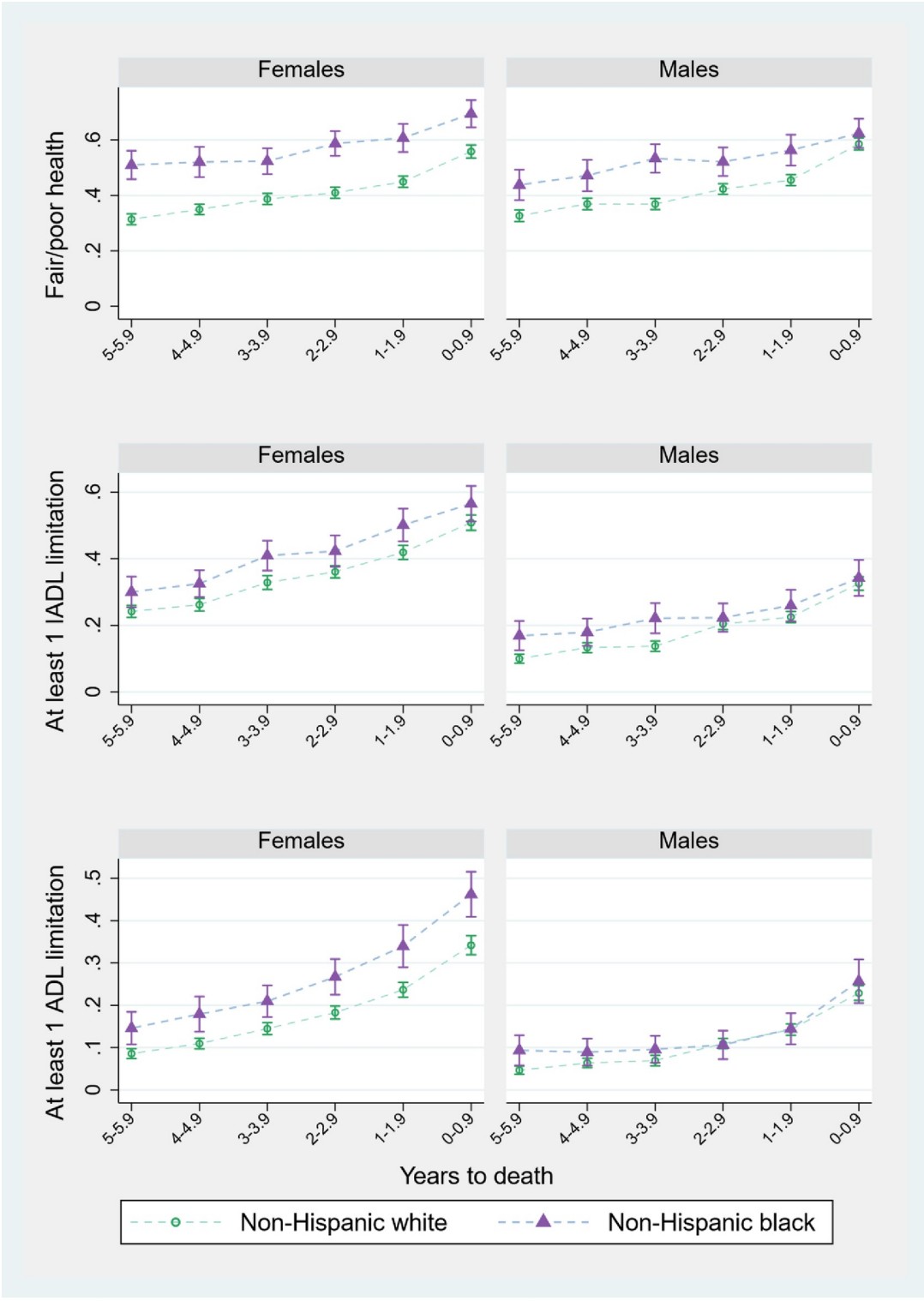

*Source*: Self-rated health from NHIS 1987-2014, IADL and ADL from NHIS 1997-2014.

**Fig 3. Proportion of population in given health state across final years of life, by race (with 95% confidence intervals).**

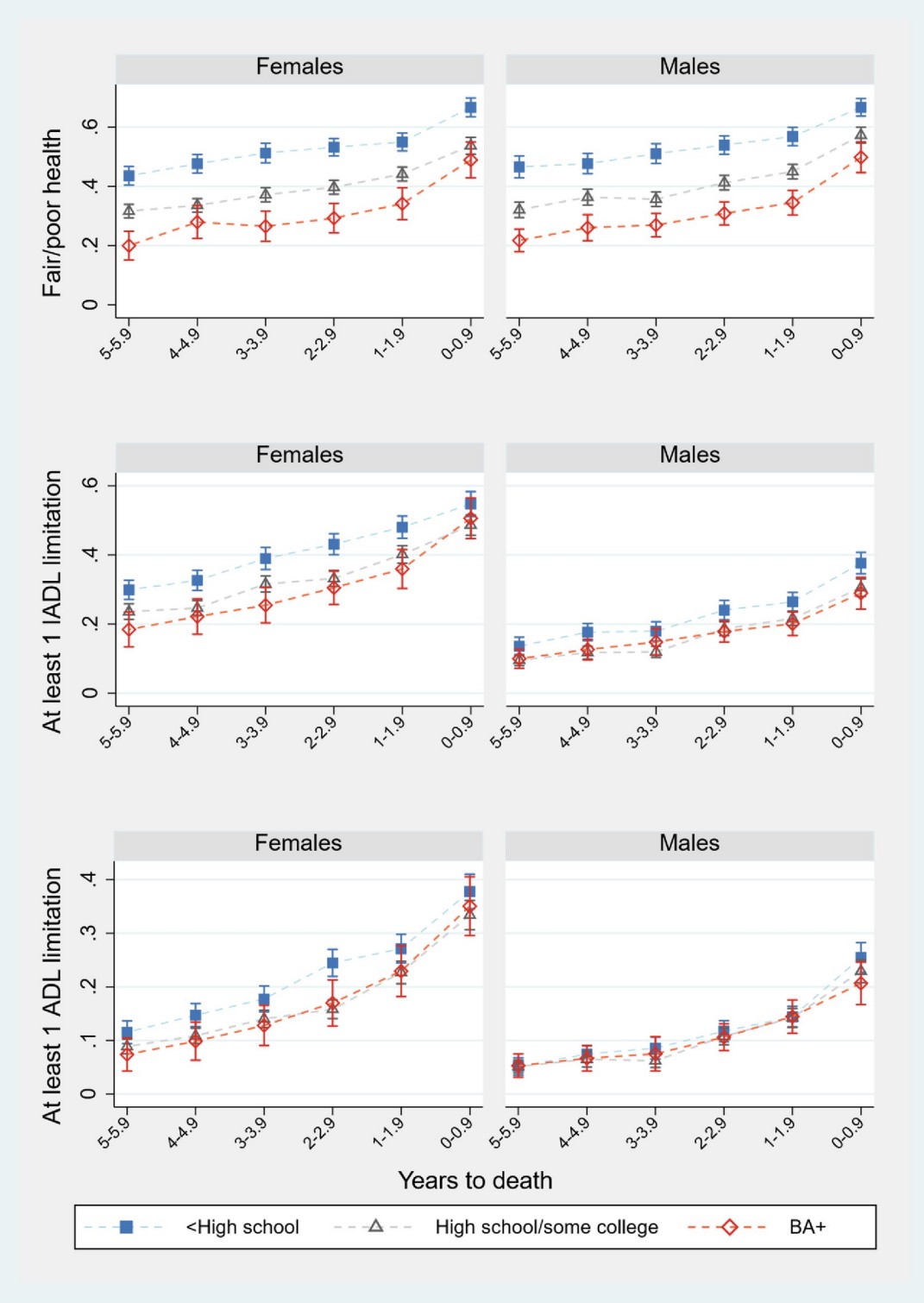

*Source*: Self-rated health from NHIS 1987-2014, IADL and ADL from NHIS 1997-2014

**Fig 4. Proportion of population in given health state across final years of life, by educational attainment (with 95% confidence intervals).**

preceding death, more than doubling for most outcomes. The trajectory of worsening SRH is remarkably similar across ages at death, with about half of the last year of life lived in adverse health (also interpretable as half of the population reporting adverse SRH in the last year). Regardless of age, most adults are similarly disabled six years before death (with the exception of older women requiring more IADL assistance than younger women). However, in contrast to SRH, a distinct age pattern emerges as death draws closer. While disability trajectories look similar for those dying between 65–74 and 75–84 years-old, the decline is much steeper for the oldest men and women. In their final year of life, more than 45% of women who survive to age 85 require help with at least one basic care task such as bathing or walking, compared to ~30% of women who die at earlier ages. Over half of the age difference in time spent with an ADL limitation in Fig 2 is due to older decedents being more likely to have 3+ limitations (S2 Appendix). Nevertheless, older decedents report being in better health for slightly longer. While the oldest women spend 8 and 14 months longer with an IADL and IADL limitation than the average of their younger counterparts, they report 4 more months of good-to-excellent health. Similarly, the oldest men experience 4–6 more disabled months than younger men, but 5 months more of favorable health (S2 Appendix).

Fig 3 compares trajectories for decedents who identify as non-Hispanic White and non-Hispanic Black. Black-White differences in the last six years of life are most apparent among women, especially in the proportion reporting fair/poor health. Over half of Black women report being in fair/poor health six years before death, compared to a third of White women. Of all population subgroups listed in S2 Appendix, Black women spend the longest time in fair/poor health (3.44 years), nearly 1 year longer than White women (2.47). Even though Black women die at younger ages, ages which are typically associated with shorter periods of disability, Black women also spend the most time with severe disability. On average, Black women require help with at least three basic care tasks, like eating or bathing, for a full year before their death (compared to 7 months for White women). These differences persist into the last year of life: 70% of Black women report fair/poor health one year before death, compared to 55% of White women. Racial differences in total time spent in adverse health are about half the size for men and decrease to statistical non-significance in the last year of life.

Fig 4 shows results by educational attainment. Educational gradients in end-of-life SRH are more pronounced than differences in disability. Even six years before death, the three education groups (<high school, high school or some college, bachelor's degree or more) report distinct levels of health, with high school dropouts being twice as likely to report adverse health than college graduates. The differentials remain constant over the next 6 years. Altogether, men and women with a college degree enjoy 16 more months of good-excellent health in the final 6 years than high school dropouts (S2 Appendix). For limitations, the gradients are smaller, or even nonexistent—though adults without a high school degree, particularly women without a high school degree, are an outlying group that is more likely than others to report a disability. As with Black-White differences in disability among men, educational differences among men in IADL limitations are small and nonexistent in ADL limitations.

## Discussion

Despite concerns about expanding morbidity at the end of life, I find that the amount of time individuals report unfavorable health in the last six years of life declined two months from 1987–2008. To the author's knowledge, this is the first study to examine trends in SRH at the end of life. I also find no change in the length of time spent with at least one end-of-life IADL or ADL limitation from 1997–2008, barring a slight increase in the most recent period. These findings are generally consistent with work by Smith et al. [12], who document an unchanging

prevalence of ADL limitations from 1995–2009 in the last 2 years of life. While Cutler et al. [14] use repeated cross sections of the Medicare Current Beneficiaries Survey linked to death records from 1991–2009 to document a decline in the prevalence of ADL and IADL limitations in the last 5 years of life in the 1990's, they find no significant change in the following decade (the main focus of this analysis). My findings are in contrast to those by Beltrán-Sánchez et al. [13], who consider six major chronic conditions and find that the adult disease burden may have grown over a similar period. Perhaps our findings differ because of the operationalization of disability versus chronic conditions. A growing disease burden might not translate to a higher prevalence of reporting one or more limitation, especially if the increases in chronic conditions are among people who already have at least one disability.

I find that the presence of disability does not always translate into a perception of poor health. Although respondents who reach ages 85+ are more likely to be disabled at the end of life than those who die before age 85 (a finding similar to the one described by Smith et al. [12]), older decedents are slightly more likely to report being in good health in their final years. Stenholm et al. [8] find a similar pattern in the Health and Retirement Survey. One reason for this paradox may be that a perception of health is constructed by comparisons with reference to individuals of a similar age [32].

Another illustration of the disconnect between disability and perceived health at the end of life is sex differences therein. It is well-documented that women report worse health than men at the same ages because women are more likely to develop disabling conditions [33]. Even at the end of life, women in the present analysis spend twice as long as men living with an IADL limitation and 70% more time with an ADL limitation in the final six years. Surprisingly, sex differences in SRH at the end of life do not reflect the expected pattern. Both sexes are equally likely to report fair/poor health at the end of life, even though women report considerably more disability. The mechanisms through which death levels sex differences in SRH, but not disability, warrant further investigation.

In contrast to the small differences in SRH but sizeable differences in disability by age, the opposite is true for racial and educational gradients. Especially among men, racial and educational differences in disability are relatively small six years before death, while differences in SRH are sizable. As men approach death, these gradients become smaller or even converge in the final 1–2 years of life, consistent with findings by Lunney et al. [15]. While disparities also disappear for women in Lunney et. al's study, Black women and women with lower levels of formal education in the present study consistently spend more time in fair/poor health and with a disability, even at the very end of life. Lunney et al.'s study population has a different age structure and is limited to adults living in Memphis and Pittsburgh. Perhaps the end-of-life experiences of women living in these two cities are not entirely representative of patterns among women nation-wide. While past work suggests that racial differences in some outcomes are largely explained by racial differences in educational attainment and socio-economic status [11,27], the explanation remains unsatisfactory for women [27]. Future work should consider other, traditionally non-measured factors, like stress and discrimination [34,35], as well as the intersection of racism, sexism, and ageism.

The implications of health decline at the end of life extend to other generations. Informal caregiving provided by relatives is the most common form of elder care [17]. Black individuals are more likely to be caregivers for family members, to spend more time caregiving, and to care for a family member with 3+ ADL limitations [36]. These facts are consistent with the findings of the present analysis. Black adults (particularly women) not only require care for longer, but require more intensive care. I find that the majority of racial disparities in one or more ADL limitations is driven by adults reporting three or more disabilities (S2 Appendix). Future work should examine which chronic conditions drive these racial disparities in higher

order disability. Furthermore, given the unequal distribution of the financial, physical, and emotional burden of prolonged caregiving [37], future research should also consider the extent to which informal caregiving of family members is a vehicle for the intergenerational transmission of inequality.

## Limitations

The NHIS does not interview individuals who at the time of interview live in a long-term care facility, though respondents who enter a care facility at some point after their NHIS interview are still linked to their death certificate. In other words, a non-institutionalized respondent can be interviewed by the NHIS before moving into long-term care some time later and remain in the follow-up group. The effects of excluding institutionalized individuals from the baseline interview are likely minimized by the typically short duration of residence in end-of-life care facilities [23,38]. The median length of stay for nursing home residents is 5 months, and 53% of residents die within 6 months of admission. The median stay is longer for women (8 vs. 3 months for men), Black individuals (6.5 vs. 5 months for Whites), and poorer groups (9 months for the bottom income quartile vs. 3 for the top) [38]. Since the sickest members of these groups are more likely to be institutionalized at baseline, and therefore excluded from the sample, between-group estimates are likely conservative. Evidence of convergence in disparities in end-of-life health in this analysis should therefore be interpreted with caution.

Another bias to consider when comparing differences across groups is selective mortality. Since the analysis only includes adults who survive to age 65, the estimates of between-group differences are likely smaller than they would be if mortality before age 65 were random.

A limitation of repeated cross-sectional data is that it is not possible to distinguish between period and cohort effects. Measuring the relative importance of cohort composition versus period changes in the treatment of illness and disability would help target relevant interventions. For example, knowing that older male cohorts are less likely than their predecessors to be heavy smokers [19], but more likely to face the health problems associated with obesity [39], could inform the decision to divert funds away from tobacco-related interventions and toward new programs targeting obesity.

## Conclusion

I report three main findings. First, despite rising ages at death, the findings indicate that the period of poor health and disability prior to death has not been extended in recent years. In this analysis, time in unfavorable health in the last six years of life declined by two months from 1987–2008, and time spent with at least one activity limitation from 1997–2008 remained stable. Second, even though older decedents and women are disabled for longer at the end of life, they report similar health to that of younger decedents and males, respectively. This paradox stands in contrast to well-studied sex differences at older chronological ages at which women report worse health and more disability. Third, the cross-sectional data suggests that while death reduces or even equalizes all racial and educational disparities among men, inequalities in healthy aging at the end of life persist for women.

Previous work finds that unequal access to formal education has a significant influence on end-of-life inequalities [11,27] and adds another item to the long list of benefits to expanding educational access. Minimizing disparities in educational outcomes is a long-term approach to reducing disparities at the end of life. In the meantime, the symptoms of health inequality could be addressed with additional support for those who report longer periods of ill health before death. In particular, women in this analysis require 6–12 more months of help with a limitation than men (for a total of 14 months lived with an ADL and 26 months with an IADL

limitation). Similarly, Black women and women with less than a high school education require assistance for longer and report prolonged periods of unfavorable health. Ensuring access to and knowledge about programs that pay family members, including spouses, to act as a caregiver in affected communities may be beneficial. Support groups, including telephone hotlines, might be sources of social support for aging adults and caregivers. Finally, given the significant racial differences in reporting multiple limitations, future work should explore disparities in specific chronic conditions to create targeted interventions.

## Supporting information

**S1 Appendix. Years out of last six years of life spent in each health state for decedents 65+, over time.**
(DOCX)

**S2 Appendix. Years out of last six years of life spent in each health state for decedents 65+, by age at death, race, and educational attainment.**
(DOCX)

## Author Contributions

**Conceptualization:** Yana C. Vierboom.

**Investigation:** Yana C. Vierboom.

**Methodology:** Yana C. Vierboom.

**Software:** Yana C. Vierboom.

**Validation:** Yana C. Vierboom.

**Writing – original draft:** Yana C. Vierboom.

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
