## [Decision Letter · Decision Letter 0]

25 Jun 2021

PONE-D-21-16309

How does it all end?

Trends and disparities in health at the end of life

PLOS ONE

Dear Dr. Vierboom,

Thank you for submitting your manuscript to PLOS ONE. After careful consideration, we feel that it has merit but does not fully meet PLOS ONE’s publication criteria as it currently stands. Therefore, we invite you to submit a revised version of the manuscript that addresses the points raised during the review process.

The two reviewers, both experts in the areas of health, aging, and end of life research, have given careful consideration to your manuscript.  Please address each of their thoughtful comments in your response letter.  Both reviewers would like to see a better contextualization of the study in the existing literature, and more justification of some of your analytic decisions, for example: Why 6 years? Why these cause of death groupings?  In addition, please address further the important limitation of the NHIS exclusion of the institutionalized population, and the selection effect that presents.  Please explain what is meant in the Limitations section where it says that the newly-institutionalized NHIS respondents "remain in the sample" since the NHIS is cross-sectional.  Also, please explain. the decision to exclude Hispanics and limit comparisons to non-Hispanic whites and Blacks.

We look forward to receiving your revised manuscript.

Kind regards,

Ellen L. Idler

Academic Editor

PLOS ONE

Journal Requirements:

Additional Editor Comments (if provided):

Reviewers' comments:

Reviewer's Responses to Questions

**Comments to the Author**

1. Is the manuscript technically sound, and do the data support the conclusions?

Reviewer #1: Partly

Reviewer #2: Partly

2. Has the statistical analysis been performed appropriately and rigorously? 

Reviewer #1: Yes

Reviewer #2: Yes

3. Have the authors made all data underlying the findings in their manuscript fully available?

Reviewer #1: Yes

Reviewer #2: Yes

4. Is the manuscript presented in an intelligible fashion and written in standard English?

Reviewer #1: Yes

Reviewer #2: Yes

5. Review Comments to the Author

Reviewer #1: I indicate above that the statistical analysis has been preformed appropriately and rigorously. However, I raise concerns about how certain variables are operationalized and indicate some analysis that might be missing. Based on the analysis plan reported by the authors (and I indicate is lacking), analysis appears to have been performed appropriately and rigorously. Please see attached file for additional comments.

Reviewer #2: This well-written descriptive paper provides a detailed statistical portrait of health and disability at the end of life, with careful attention to differences therein on the basis of sociodemographics (age, sex, race, education) and cause of death. The analyses are carefully done and use excellent NHIS data. Despite these strengths, the author could do more to justify their study aims and their key analytic decisions. I also encourage the author to say more about the possible influences of age vs. cohort effects, and selective survival when interpreting their results. I hope these comments are helpful to the author as they further develop this important project.

1. Why does the analysis focus on the final six years of life? Please provide a brief rationale for this decision. Other time points, such as last year of life, may align more closely with the literature on end-of-life medical expenditures, for instance. This shorter observation would help the author to better contextualize this in the literature, and link their findings to topics like expenditures. I suspect that the six-year decision was driven by sample size, but there may be other more compelling motivators.

2. More generally, the author could make a much more compelling case for the study goals. Why is this descriptive analysis helpful? What does it tell us, and how can these results advance research and policy/practice on end-of-life care?

3. PLOS is for a general readership, so more context is needed for readers who are not specialists in end-of-life topics. I would suggest a brief discussion of the strengths and weaknesses of using time-to-death measures to characterize end-of-life, and the relative strengths and weaknesses relative to measures like proxy reports and expenditure data that are other ways to characterize the end-of-life experience.

4. The section on inequalities in healthy aging, again, could do more to convey the importance and value of the research. This might be an opportune place to discuss the possibility of selective survival, such that lower SES and Blacks who survive until age 65 may show some health advantages relatives to higher SES and Whites who survive. This might be another interpretation of the seemingly counter-intuitive results whereby the least educated group appear to fare better than their more educated counterparts.

5. I would suggest using a more nuanced measure of age in the analysis, such as 65-74, 75-84, 85+. The current two category measure is quite coarse and some nuanced patterns may be concealed.

6. A minor issue. On page 13, the phrase “perceived amount of time individuals spend in unfavorable health…” is misleading. Study participants were not administered perceived life span or perceived probability of survival questions. Please re-word so that the sentence more accurately characterizes the data.

7. The analyses should raise issues of age versus cohort effects (and even touch on period effects) more directly, in the background and discussion. Even though the time period is fairly narrow, the youngest participants in 2014 and the oldest participants in 1987 represent vastly different cohorts, who have had different exposures to medications, public health interventions, assistive devices, etc. over the life course. At the very least, the limitations could say more about the inability to discern age vs cohort effects using repeated cross-sectional data.

Best of luck with your revision.

6. PLOS authors have the option to publish the peer review history of their article (what does this mean?). If published, this will include your full peer review and any attached files.

Reviewer #1: No

Reviewer #2: No

---

## [Author Response · Author response to Decision Letter 0]

16 Dec 2021

Dear Professor Idler and Reviewers,

Thank you for your thorough examination of my manuscript. Your thoughtful feedback has helped me draft what I think is a stronger and more precise manuscript.

I outline the biggest changes to the analysis and the text below, before listing and responding to each reviewer’s comments. My responses in the Word version of this response document are written in blue and different font. To clearly distinguish my responses in the online version of these comments, I also use AR (for “author’s response”) before my reply to each comment. Revisions to the manuscript itself are recorded in track changes in the revised manuscript file. 

Significant changes to the analysis:

1. Cause-of-death (COD) analysis: I have removed the cause-of-death analysis, in part due to Reviewer 1’s concerns about its usefulness and limited insights. I believe the purpose and scope of the manuscript are clearer without it. The analysis considers three health outcomes (SRH, IADL, ADL) stratified by age, race, and education—by sex. Previously, the analysis was also stratified by COD. Since COD is not a population characteristic as age, race, sex, and education are, it ultimately was out of place.

2. Age categories: There are now three age categories (65-74, 75-79, 85+), instead of two (Reviewer 2).

3. Educational attainment: I combined the categories of “high school” and “some college” since previous results for this older population were similar and muddled some of the patterns. 

4. Table and Figures: The original Table 1 (distribution of characteristics) has been updated with the adjusted age and education variables. I switched Table 2 (health outcomes for subgroups by time to death) with the Figures (graphical representations of health outcomes) from the Appendix, since the Figures tell the story more succinctly than the Table.

Significant changes to the text include 1) the expansion of the background section to provide more context and better situate the present study; 2) the revision of the results section to better describe findings; and 3) the expansion of the discussion and conclusion sections to more clearly illustrate the paper’s contributions and implications.

I am hopeful that these revisions will satisfy the reviewers’ concerns about the original manuscript. Thank you for considering my manuscript for publication in Plos One. 

Comments from the Editor: 

Both reviewers would like to see a better contextualization of the study in the existing literature, and more justification of some of your analytic decisions, for example: Why 6 years? Why these cause of death groupings? In addition, please address further the important limitation of the NHIS exclusion of the institutionalized population, and the selection effect that presents. Please explain what is meant in the Limitations section where it says that the newly-institutionalized NHIS respondents "remain in the sample" since the NHIS is cross-sectional. Also, please explain. the decision to exclude Hispanics and limit comparisons to non-Hispanic whites and Blacks.

AR: Thank you for this summary. I have clarified the decision to exclude non-white and non-Black respondents in the methods section: 

“Due to small sample sizes for non-Black and non-white racial/ethnic categorizations, I limit the racially-stratified analyses to Black-white comparisons. However, the other comparisons, such as by educational attainment, include all respondents, regardless of reported race/ethnicity.”

I have also clarified the language around institutionalization:

“The NHIS does not interview individuals who at the time of interview live in a long-term care facility, though respondents who enter a care facility at some point after their NHIS interview are still linked to their death certificate at death. In other words, a non-institutionalized respondent can be interviewed by the NHIS before moving into long-term care some time later and remain in the follow-up group...”

Your other points are addressed in my response letter below.

Reviewer #1:

Thank you for the opportunity to review this research article that considers trends and disparities in quality of life prior to death among adults age 65 and older in the U.S. The author investigates self-rated health (SRH) and disability in the last six years of life across: 1) time, 2) population subgroups (i.e., Black-White and male-female), and 3) causes of death. The author uses data from the 1987-2014 National Health Interview Survey, linked to death records through 2015, for respondents who died within six years of being interviewed. Decedents are classified as having a disability prior to death based on needing help with at least one of six ADLs and/or needing help with routine needs, such as grocery shopping or doing light chores. Causes of death are classified as: 1) accidents, 2) cancers, 3) cerebrovascular diseases, 4) chronic lower respiratory diseases, 5) a combination of heart disease and diabetes, and 6) all other causes. The manuscript is fairly well written and addresses an important topic (i.e., trends and disparities in health at end of life). Findings from such a study have the potential to inform interventions and guide program planning and address disparities. The author indicates that this study is the first known study to investigate trends in SRH in the last years of life which can make an important contribution to the literature. Some concerns noted below limit the potential impact of the paper.

1. The author should be more clear regarding data that are used. The abstract and Findings refer to data from the 1987-2014 National Health Interview Survey (NHIS) whereas the Methods (p. 5) refer to data from the 1997-2014 NHIS. 

AR: Thank you for pointing out this inconsistency. I use data from 1986-2014 for SRH, and 1997-2014 for limitations (since data is not available for these variables for earlier years). I have now clarified this throughout the manuscript and most explicitly in the methods section:

“… The NHIS began consistently asking respondents about ADL and IADL limitations in 1997, but has included a question on SRH every year since the 1970’s. Since respondent matching to death records did not begin until 1986, however, and because a different weighting scheme was used in 1986, I begin the analysis of SRH trends in 1987. The analysis of ADL and IADL measures begins in 1997. I examine all outcomes until 2014, the last year for which interviewed respondents in the public-use files have been linked to death records (through 2015).”

2. The rationale for using a timeframe of up to six years-to-death is unclear. How was the six years-to-death timeframe determined? 

AR: This point was also made by Reviewer #2. A longer timeframe provides more time to observe differences between groups. While a window longer than six years would have been even better, a choice of six years keeps the time trend portion of the analysis from being too outdated (since each year requires the full follow-up time to have elapsed). I have added a discussion of this balance to the Methods section:

“I assign each decedent a value for years to death by subtracting the calendar year in which a respondent was interviewed from the respondent’s year of death reported in the linked mortality file. The maximum length of follow-up before death is an important analytic decision. While some of the studies cited in this paper consider the last one or two years of life (Gill et al. 2010; Liao et al. 1999; Smith et al. 2013), others extend 3-8 years before death (Beltrán-Sánchez et al. 2016; Cutler et al. 2013; Lunney et al. 2018; Raab et al. 2018; Wilson et al. 2007), and some well beyond 10 years (Alley et al. 2010; Gerstorf et al. 2013; Stenholm et al. 2014). Lunney et al.’s (2018) findings that racial disparities in disability are ‘erased’ in the last 1-1.5 years of life suggests that a period longer than 2 years before death is needed to capture evolving patterns of disparities. The authors also find that study participants who died were more disabled at the study’s baseline three years prior to death than similarly-aged peers who did not die. This suggests that the onset of disability, an important outcome in the present study, likely begins earlier than the last few years of life. 

In this paper, I consider the last six years of life. Since the annual trends portion of the analysis requires six years to have elapsed since interview and mortality data is only available through 2015, the last year for which estimates can be produced is for respondents interviewed in 2008. Although a window longer than six years would be optimal, it strikes the balance between observing outcomes and disparities for as long as possible while tracking relatively recent trends. …”

3. Given the focus on disparities among subgroups, including investigating Black-White differences, the author might consider an earlier paper by Liao et al., (1999) that also used NHIS data and can provide important context for the current paper. Using data from the 1986-1994 NHIS, linked to death records through 1995, these researchers investigated Black-White differences in disability and mortality in decedents age 50 years and older who died within 2 years of their interview. Researchers found that adjusting for educational attainment (used as a proxy for socioeconomic status) did not eliminate Black-White differences in disability and morbidity in the last years of life but did account for much of the difference. The current paper considers differences in health status with regard to race as well as education and gender but does not adjust for any potential confounding by socioeconomic status or other social determinants with regard to Black-White differences which might be important in informing interventions to address modifiable factors associated with these determinants. 

AR: Thank you for this excellent suggestion. I now cite this important paper throughout the manuscript. As you suggest, the authors findings are also helpful for addressing modifiable factors. I discuss this in the discussion:

“Previous work finds that unequal access to formal education has a significant influence on end-of-life inequalities (Liao et al. 1999; Warner & Brown 2011) and adds another item to the long list of benefits to expanding educational access. Minimizing disparities in educational outcomes is a long-term approach to reducing disparities at the end of life. …”

4. A seminal paper by Lunney et al. (2003) that also is not cited by the authors investigated differences in functional decline among four types of illness trajectories: 1) sudden death, 2) cancer death, 3) death from organ failure, and 4) frailty. This research found that patterns of functional decline or disability varied substantially based on illness trajectory. Similar to the current paper, decedents who died suddenly or who died from cancer experienced the shortest periods of disability prior to death. Given that variations in the shape of disability trajectories by disease type is well established, findings related to cause of death in the current paper are not surprising or particularly compelling. Examining differences related to number of chronic conditions and their disabling effects at end of life might be more informative, especially since some evidence indicates that these disabling effects (i.e., based on number of chronic conditions) seem to be similar for Blacks and Whites who are approaching death. 

AR: This comment and several of your following points prompted me to reexamine the usefulness of the manuscript’s cause-of-death analysis, ultimately choosing to eliminate it. A significant aim of the analysis is to understand end-of-life trajectories for certain subpopulations. While the other stratifying characteristics (age, sex, race, educational attainment) are about membership in certain social groups, cause of death is of course difficult to know before death. I believe the analysis is more focused without this excursion and have adjusted the manuscript and tables as necessary.

The idea to examine groups by number of conditions is compelling in light of the finding that racial disparities at the end-of-life are driven by higher order limitations. I’ve noted this in the discussion section: 

“Black adults (particularly women) not only require care for longer, but require more intensive care. I find that the majority of racial disparities in one or more ADL limitations is driven by adults reporting three or more disabilities (Appendix Table 2). Future work should examine which chronic conditions are drive these racial disparities in higher order disability.”

5. Related to # 4 above, a large percentage of the sample (43% of females and 51% of males) died from accidents and cancers (shown in previous studies to be associated with the shortest periods of disability prior to death) and 41% of females and 35% of males are classified as “all other” causes of death. Only about 16% of females and 14% of males are grouped into specific non-cancer disease categories. It also is unclear why decedents with heart disease and diabetes are grouped into one category, even if people with diabetes are at high risk for heart disease. These cause of death categories do not seem particularly meaningful or informative. 

AR: Please see response to comment #4.

6. Related to #4 and #5 above, the author does not indicate how overlaps in chronic conditions are handled. Given that multiple chronic conditions are common among people age 65 and older, it is likely that many decedents reported more than one chronic condition.

AR: Please see response to comment #4.

7. Related to #4, #5, and #6 above, the Liao et al., (1999) paper (referenced above under #3) that also used NHIS data to investigate disparities and trends in disability in the last years of life excluded decedents who died from accidental causes. The author might want to also consider excluding these decedents. They do not seem theoretically relevant. It also would be useful for the author to better describe how information about chronic conditions is collected in the NHIS. What are the range and types of conditions reflected in the survey?

AR: Please see response to comment #4. I do not use any data on chronic conditions. Rather, I use only data on reported difficulty with IADL or ADLs. 

8. On p. 15, the authors indicates not including Alzheimer’s disease as a cause of death in the analysis based on the high likelihood of institutionalization with disease progression. However, the author indicates on p. 15 that decedents who enter a care facility after their interview are retained in the sample. Again, the rationale for the cause of death categories used in this study is questionable. 

AR: Please see response to comment #4.

9. In addition, it would be useful to have more information regarding the measure of IADLs used in this study beyond just grocery shopping or doing light chores. Help with grocery shopping can be related to a lack of transportation as opposed to disability. The six ADLs are listed but less information is provided about the IADL measure. 

AR: Unfortunately, the NHIS does not include additional information as to the IADL task with which a respondent requires help. Rather, the IADL measure is a simple yes/no question about whether a respondent requires any help with any IADL. I have clarified this in the Methods section:

“… The NHIS ascertained information on IADL limitations in the NHIS using a single yes/no question for whether a respondent needed help from others for ‘handling routine needs, such as everyday household chores, doing necessary business, shopping, or getting around for other purposes.’ While the above ADL variable combines six survey questions, the IADL variable reflects only this single question.”

Reviewer #2: This well-written descriptive paper provides a detailed statistical portrait of health and disability at the end of life, with careful attention to differences therein on the basis of sociodemographics (age, sex, race, education) and cause of death. The analyses are carefully done and use excellent NHIS data. Despite these strengths, the author could do more to justify their study aims and their key analytic decisions. I also encourage the author to say more about the possible influences of age vs. cohort effects, and selective survival when interpreting their results. I hope these comments are helpful to the author as they further develop this important project.

1. Why does the analysis focus on the final six years of life? Please provide a brief rationale for this decision. Other time points, such as last year of life, may align more closely with the literature on end-of-life medical expenditures, for instance. This shorter observation would help the author to better contextualize this in the literature, and link their findings to topics like expenditures. I suspect that the six-year decision was driven by sample size, but there may be other more compelling motivators.

AR: Thank you. A similar point was made by Reviewer #1. Please see my response to their comment #2 above. 

2. More generally, the author could make a much more compelling case for the study goals. Why is this descriptive analysis helpful? What does it tell us, and how can these results advance research and policy/practice on end-of-life care?

AR: This point was echoed by the Editor. I have significantly rewritten parts of the introduction, discussion, and conclusion to better highlight my findings and make clearer recommendations for future research and policies.

3. PLOS is for a general readership, so more context is needed for readers who are not specialists in end-of-life topics. I would suggest a brief discussion of the strengths and weaknesses of using time-to-death measures to characterize end-of-life, and the relative strengths and weaknesses relative to measures like proxy reports and expenditure data that are other ways to characterize the end-of-life experience.

AR: Thank you for the helpful suggestion. I have rewritten part of the background section:

“From evaluating the financial wellbeing of pension systems to predicting a population’s healthcare needs, the end-of-life period is of interest across disciplines. A significant analytic decision in end-of-life studies is whether to measure age since birth, as is most typical, or to measure backward from the other end of the lifespan: death. The usefulness of a variable for remaining lifetime was first described in the 1970’s (see Sanderson & Scherbov 2013 for a history of the variable, as well as a demonstration of using the variable to study population aging). Years to death is a proxy for the complex and interacting social, behavioral, environmental, and genetic processes that determine each individual’s moment of death. An allure of the variable is that it is still under-explored, despite yielding new perspectives that are missed when using only chronological age.”

4. The section on inequalities in healthy aging, again, could do more to convey the importance and value of the research. This might be an opportune place to discuss the possibility of selective survival, such that lower SES and Blacks who survive until age 65 may show some health advantages relatives to higher SES and Whites who survive. This might be another interpretation of the seemingly counter-intuitive results whereby the least educated group appear to fare better than their more educated counterparts.

AR: Discussing selective mortality is a very helpful suggestion. I now do this in the inequalities section as you suggest:

“Another possibility through which inequalities may diminish is selective mortality. Since some populations are exposed to systematically higher mortality rates throughout their lives, these groups can be highly select by the time they reach the ages under study. By nature of their design, studies using chronological age (comparing 80 year-old white adults to their 80 year-old Black peers, for example) must ignore the influence of selective mortality. The issue is greatly lessened when considering time-to-death (comparing racial differences five years before death, for example), but nevertheless persists anytime a study sample has a minimum age below which differential mortality occurs.”

I also reference it again in the section on limitations.

“Another bias to consider when comparing differences across groups is selective mortality. Since the analysis only includes adults who survive to age 65, the estimates of between-group differences are likely smaller than they would be if everyone survived to age 65.”

5. I would suggest using a more nuanced measure of age in the analysis, such as 65-74, 75-84, 85+. The current two category measure is quite coarse and some nuanced patterns may be concealed.

AR: Thank you for the suggestion. I have updated the age categories to the ones you suggest. The new age ranges do indeed offer a more nuanced understanding and I have updated the manuscript accordingly.

6. A minor issue. On page 13, the phrase “perceived amount of time individuals spend in unfavorable health…” is misleading. Study participants were not administered perceived life span or perceived probability of survival questions. Please re-word so that the sentence more accurately characterizes the data.

AR: Good point. I have removed the word “perceived” and checked that similar phrasing is not used throughout the manuscript.

7. The analyses should raise issues of age versus cohort effects (and even touch on period effects) more directly, in the background and discussion. Even though the time period is fairly narrow, the youngest participants in 2014 and the oldest participants in 1987 represent vastly different cohorts, who have had different exposures to medications, public health interventions, assistive devices, etc. over the life course. At the very least, the limitations could say more about the inability to discern age vs cohort effects using repeated cross-sectional data.

AR: Thank you for this excellent point. In the background section where I discuss trends, I touch on changing population characteristics and environments. I now explicitly link this discussion to period vs. cohort effects:

“Because of these simultaneously evolving environments and population characteristics, it is difficult to separate period and cohort effects.”

I now also refer to the issue again in the limitations section:

“Finally, a limitation of repeated cross-sectional data is that it is not possible to distinguish between period and cohort effects. Measuring the relative importance of cohort composition vs. period changes in the treatment of illness and disability would help target relevant interventions.”

---

## [Decision Letter · Decision Letter 1]

2 Feb 2022

PONE-D-21-16309R1How does it all end?

Trends and disparities in health at the end of lifePLOS ONE

Dear Dr. Vierboom,

Thank you for submitting your manuscript to PLOS ONE. After careful consideration, we feel that it has merit but does not fully meet PLOS ONE’s publication criteria as it currently stands. Therefore, we invite you to submit a revised version of the manuscript that addresses the points raised during the review process.

Both reviewers from the original round have reviewed the paper again and both see substantial improvement in the way the study is framed and carried out.  However, both reviewers would like to see additional improvements in the writing, to bring the paper to a higher level of quality.  They have made specific suggestions and provided examples that will be helpful.  

We look forward to receiving your revised manuscript.

Kind regards,

Ellen L. Idler

Academic Editor

PLOS ONE

Journal Requirements:

Reviewers' comments:

Reviewer's Responses to Questions

**Comments to the Author**

1. If the authors have adequately addressed your comments raised in a previous round of review and you feel that this manuscript is now acceptable for publication, you may indicate that here to bypass the “Comments to the Author” section, enter your conflict of interest statement in the “Confidential to Editor” section, and submit your "Accept" recommendation.

Reviewer #1: (No Response)

Reviewer #2: All comments have been addressed

2. Is the manuscript technically sound, and do the data support the conclusions?

Reviewer #1: Yes

Reviewer #2: Yes

3. Has the statistical analysis been performed appropriately and rigorously? 

Reviewer #1: Yes

Reviewer #2: Yes

4. Have the authors made all data underlying the findings in their manuscript fully available?

Reviewer #1: Yes

Reviewer #2: Yes

5. Is the manuscript presented in an intelligible fashion and written in standard English?

Reviewer #1: No

Reviewer #2: Yes

6. Review Comments to the Author

Reviewer #1: Thank you for the opportunity to review this revised manuscript that considers trends and disparities in end-of-life health in the U.S. The author made a good effort to address both reviewers’ (and the editor’s) comments/concerns and has strengthened the paper. Below, I have outlined some additional points to consider. Some additional editing and wordsmithing are needed and would further improve the quality of the paper.

1. The abstract could use some editing. For example, the sentence: “Time spent in fair/poor health over years 1987-2008 declined by two months, while time lived with at least one activity limitation generally remained stable from 1997-2008” is a little clunky. Time spent in fair/poor health over years 1987-2008 declined by two months compared to what? Declined by two months each year? The sentences that follow also could use some editing. The sentence: “Compared to men, women reported an IADL for 1 year longer and an ADL for 8 extra months, yet both sexes reported similar lengths of unfavorable health” also needs editing. The numbers 1 and 8 should be spelled out. Reported an IADL or ADL what? Limitation? what does the author mean by “similar lengths of unfavorable health? The way that the sentence is currently written, it is not clear what is meant here. In the next sentence, I suggest saying similar health “compared with” younger decedents instead of “to” younger decedents. Given the use of repeated cross-sectional data, I suggest using language such as “findings indicate” rather than stating the findings as if they are fact in the Discussion section of the Abstract.

2. On page 4, the author indicates the importance of examining ADL limitations (e.g., 40% of people over age 65 have at least one limitation and nearly 90% with 3+ limitations require caregiving help) but do not say why it might be important to also consider IADL limitations. They do make the case on p. 10 and may want to say something similar here on p. 4 as well. The sentence: “ADL limitations are predictive of requiring physical assistance, with roughly 40% of community-dwelling adults age 65+ with one limitation and nearly 90% with 3+ receiving caregiving help” needs editing to read more clearly. Also, physical assistance is caregiving help.

3. On p. 5, what is meant by the sentence: “Because of these simultaneously evolving environments and population characteristics, it is difficult to separate period and cohort effects.” I am not following this line of thought. I agree that the methods used in this paper (and acknowledged under Limitations) limit the ability to disentangle these effects.

4. On p. 7, instead of using wording such as, “an older study similarly constructed to this one,” consider, for example, referring to “a similar study” or “another study with a similar design,” etc. The wording here is clunky.

5. In describing the sample on p. 9 in the Methods section, information should be included regarding any exclusion criteria (e.g., living in a long-term care facility at baseline). Information should also be included here regarding the possibility that those not in long-term care at baseline could be included at follow up.

6. The literature reported in the first paragraph under the section on "Years-to death” on p. 9 in the Methods section feels out of place here. This literature should be woven into the Background section that begins on p. 3 and not here. A Methods section typically does not include a literature review. The rationale for the six years-to death timeframe can be made without including all of these citations in the Methods section.

7. On p. 12, I suggest more clearly outlining how data were pooled to conduct the varying analyses and describing the different analytic approaches in turn rather than stating: “To compare estimates across population subgroups, I pool data across years 1997-2014 (1987-2014 for SRH).”

8. Avoid using language like, “women reported either kinds of limitations for much longer” on p. 14. Again, the paper could benefit from some wordsmithing throughout to improve readability and flow and to sound more academic.

9. On p. 21, avoid using the term “elder.” The term, “older adults” is preferred by aging researchers and others.

10. When referring to Blacks and Whites, I suggest capitalizing both terms to be consistent.

11. Throughout the body of the text, “vs.” should be written out as “versus” unless included in material with parentheses.

12. On p. 11, instead of “non-Black and non-white racial/ethnic categorizations,” I might indicate that sample sizes for other racial and ethnic groups were too small to conduct any additional racial/ethnic comparisons. The language used here is a little clunky.

13. On pp. 8-11, text indicates that different sample sizes were used for different analyses (e.g., SRH vs. ADLs and IADLs as well as racial comparisons vs. other types of comparisons (e.g., educational attainment). It is not clear what the different sample sizes were for the various analyses.

14. On p. 23, The author indicates that “measuring the relative importance of cohort composition vs. period changes in the treatment of illness,” etc. would help target relevant interventions.” A brief example of how this type of analysis could inform relevant interventions might be useful here. As noted by Reviewer 2, the methods used and inability to disentangle age versus cohort versus period effects is a limitation that should be acknowledged. The author acknowledges this limitation but does not provide a concrete explanation of how/why the analysis is limited because of this constraint.

15. In the conclusion on p. 23, I suggest saying something like, “findings indicate”, etc., rather than stating results as fact.

Reviewer #2: The authors have done a very thorough and responsive revision. I have just a few remaining suggestions for improvement.

1. The manuscript would benefit from a very careful copy-edit both for clarity and style. For instance, the paper opens on an awkward note. The first sentence of a manuscript should be more direct, such as “Death—and the months, days, or moments preceding it—are an important and distinct stage of the life course (Cohen-Mansfield et al. 2018)” (deleting initial clause).

2. A recent paper by Carr and Luth (2019) Annual Review of Sociology makes the case that ‘end of life’ is a distinctive life course stage. This article could be helpful for your framing of the analysis.

3. The front end of the paper could do a bit more to motivate the selection of the multiple health measures. These issues are addressed somewhat in the Discussion, but it would be helpful to foreground the fact that SRH is subjective and assessed in relation to one’s peers (who may be dying or in poor health), whereas the IADL and ADL measures are more ‘objective’ and reflect behavioral capacities.

Overall, an interesting and creative paper that will make a nice contribution to the literature.

7. PLOS authors have the option to publish the peer review history of their article (what does this mean?). If published, this will include your full peer review and any attached files.

Reviewer #1: No

Reviewer #2: No

---

## [Author Response · Author response to Decision Letter 1]

3 Mar 2022

Response to Reviewers

Dear Professor Idler and Reviewers,

I would like to thank you again for the time you have taken to provide constructive feedback. As before, I below respond to each point with AR (for “Author’s Response”).

Best,

The Author 

Reviewer #1

Thank you for the opportunity to review this revised manuscript that considers trends and disparities in end-of-life health in the U.S. The author made a good effort to address both reviewers’ (and the editor’s) comments/concerns and has strengthened the paper. Below, I have outlined some additional points to consider. Some additional editing and wordsmithing are needed and would further improve the quality of the paper.

1. The abstract could use some editing. For example, the sentence: “Time spent in fair/poor health over years 1987-2008 declined by two months, while time lived with at least one activity limitation generally remained stable from 1997-2008” is a little clunky. Time spent in fair/poor health over years 1987-2008 declined by two months compared to what? Declined by two months each year? The sentences that follow also could use some editing. The sentence: “Compared to men, women reported an IADL for 1 year longer and an ADL for 8 extra months, yet both sexes reported similar lengths of unfavorable health” also needs editing. The numbers 1 and 8 should be spelled out. Reported an IADL or ADL what? Limitation? what does the author mean by “similar lengths of unfavorable health? The way that the sentence is currently written, it is not clear what is meant here. In the next sentence, I suggest saying similar health “compared with” younger decedents instead of “to” younger decedents. Given the use of repeated cross-sectional data, I suggest using language such as “findings indicate” rather than stating the findings as if they are fact in the Discussion section of the Abstract.

AR: Thank you for these suggestions. I have significantly re-written the Results and Discussion sections of the abstract.

2. On page 4, the author indicates the importance of examining ADL limitations (e.g., 40% of people over age 65 have at least one limitation and nearly 90% with 3+ limitations require caregiving help) but do not say why it might be important to also consider IADL limitations. They do make the case on p. 10 and may want to say something similar here on p. 4 as well. The sentence: “ADL limitations are predictive of requiring physical assistance, with roughly 40% of community-dwelling adults age 65+ with one limitation and nearly 90% with 3+ receiving caregiving help” needs editing to read more clearly. Also, physical assistance is caregiving help.

AR: In addition to clarifying the suggested sentence, I have added a sentence to expand on the utility of IADL limitations: 

“While IADL limitations are less disabling than ADL limitations, an IADL limitation indicates that an individual requires some level of support in order to live independently.”

3. On p. 5, what is meant by the sentence: “Because of these simultaneously evolving environments and population characteristics, it is difficult to separate period and cohort effects.” I am not following this line of thought. I agree that the methods used in this paper (and acknowledged under Limitations) limit the ability to disentangle these effects.

AR: I have re-written the sentence to read:

“Because cohorts are evolving at the same time as the contexts in which they live, it is difficult to separate period and cohort effects.”

4. On p. 7, instead of using wording such as, “an older study similarly constructed to this one,” consider, for example, referring to “a similar study” or “another study with a similar design,” etc. The wording here is clunky.

AR: I have made the change to “a similar study”.

5. In describing the sample on p. 9 in the Methods section, information should be included regarding any exclusion criteria (e.g., living in a long-term care facility at baseline). Information should also be included here regarding the possibility that those not in long-term care at baseline could be included at follow up.

AR: I have added the following sentence:

“While residents of long-term care institutions are not included in the baseline sample, the sample may include individuals who were interviewed at home and then moved to an institution during the follow-up period.”

6. The literature reported in the first paragraph under the section on "Years-to death” on p. 9 in the Methods section feels out of place here. This literature should be woven into the Background section that begins on p. 3 and not here. A Methods section typically does not include a literature review. The rationale for the six years-to death timeframe can be made without including all of these citations in the Methods section.

AR: As you suggest, I have moved the section with the citations to the Background section, while maintaining a shortened rationale for the time frame in the Methods section.

7. On p. 12, I suggest more clearly outlining how data were pooled to conduct the varying analyses and describing the different analytic approaches in turn rather than stating: “To compare estimates across population subgroups, I pool data across years 1997-2014 (1987-2014 for SRH).”

AR: I have rewritten this sentence and avoided using the term “pooling”, which made the approach sound unnecessarily complicated. The sentence now reads: 

“To compare outcomes across population subgroups, I estimate the mean of an outcome prevalence across the study period (years 1997-2014 for SRH; years 1997-2014 for IADL and ADL).”

8. Avoid using language like, “women reported either kinds of limitations for much longer” on p. 14. Again, the paper could benefit from some wordsmithing throughout to improve readability and flow and to sound more academic.

AR: I have adjusted the wording and gone through the manuscript to find a balance between accessibility and sounding academic.

9. On p. 21, avoid using the term “elder.” The term, “older adults” is preferred by aging researchers and others.

AR: I have the term changed “elder care” to “care for older adults.”

10. When referring to Blacks and Whites, I suggest capitalizing both terms to be consistent.

AR: Thank you for this suggestion. I gave careful thought to my decision to capitalize Black, but not white, in my past submissions of the manuscript. The stylistic debate today is of course tied to discussions around identity and racism in the US today. 

Around the time of my first submission in 2021, the Associated Press announced that it would capitalize Black, and not white. 

This thought piece in The Atlantic also gives a good overview of the current debate. 

On reconsideration of the current debate (and in particular this piece by Dr. Eve Ewing), I have chosen to capitalize White. 

The initial aim was of the typographical inconsistency was to highlight the upper case B. A prominent argument against the lowercase w, however, is that its use doesn’t recognize Whiteness as a sociological construct and allows White people to sit out on the debates and work around racism. 

Therefore, I have made the change throughout.

11. Throughout the body of the text, “vs.” should be written out as “versus” unless included in material with parentheses.

AR: Thank you. I have made two necessary changes.

12. On p. 11, instead of “non-Black and non-white racial/ethnic categorizations,” I might indicate that sample sizes for other racial and ethnic groups were too small to conduct any additional racial/ethnic comparisons. The language used here is a little clunky.

AR: Good point. The sentence now reads:

“Due to the small sample sizes for other racial/ethnic categorizations, …”

13. On pp. 8-11, text indicates that different sample sizes were used for different analyses (e.g., SRH vs. ADLs and IADLs as well as racial comparisons vs. other types of comparisons (e.g., educational attainment). It is not clear what the different sample sizes were for the various analyses.

AR: Thank you for pointing this out. The information was previously buried in the notes for Table 1. I now mention this in a footnote in the section describing the sample size:

“The SRH analysis sample consists of 77,295 individuals across all years 1987-2014. Sample sizes for the IADL and ADL analyses, which span years 1997-2014, are 40,354 and 40,359, respectively.”

14. On p. 23, The author indicates that “measuring the relative importance of cohort composition vs. period changes in the treatment of illness,” etc. would help target relevant interventions.” A brief example of how this type of analysis could inform relevant interventions might be useful here. As noted by Reviewer 2, the methods used and inability to disentangle age versus cohort versus period effects is a limitation that should be acknowledged. The author acknowledges this limitation but does not provide a concrete explanation of how/why the analysis is limited because of this constraint.

AR: Thank you for the suggestion. I have added the second sentence:

“Measuring the relative importance of cohort composition versus period changes in the treatment of illness and disability would help target relevant interventions. For example, knowing that older male cohorts are less likely than their predecessors to be heavy smokers (Preston & Wang 2006), but more likely to face the health problems associated with obesity (Wang et al 2011), could inform the decision to divert funds away from tobacco-related interventions and toward new programs targeting obesity.”

15. In the conclusion on p. 23, I suggest saying something like, “findings indicate”, etc., rather than stating results as fact.

AR: I now use weaker language throughout the discussion. 

Reviewer #2 

The authors have done a very thorough and responsive revision. I have just a few remaining suggestions for improvement.

1. The manuscript would benefit from a very careful copy-edit both for clarity and style. For instance, the paper opens on an awkward note. The first sentence of a manuscript should be more direct, such as “Death—and the months, days, or moments preceding it—are an important and distinct stage of the life course (Cohen-Mansfield et al. 2018)” (deleting initial clause).

AR: Thank you for the suggestion. I have adjusted the first sentence and have made clarifying changes throughout the manuscript.

2. A recent paper by Carr and Luth (2019) Annual Review of Sociology makes the case that ‘end of life’ is a distinctive life course stage. This article could be helpful for your framing of the analysis.

AR: I now cite this helpful paper throughout the manuscript.

3. The front end of the paper could do a bit more to motivate the selection of the multiple health measures. These issues are addressed somewhat in the Discussion, but it would be helpful to foreground the fact that SRH is subjective and assessed in relation to one’s peers (who may be dying or in poor health), whereas the IADL and ADL measures are more ‘objective’ and reflect behavioral capacities.

AR: Thank you for this suggestion. The Background section now reads:

“SRH is a subjective and self-reported indicator of health. While disability is also self-reported, it serves as a more objective measure of requiring assistance.”

---

## [Decision Letter · Decision Letter 2]

12 Apr 2022

How does it all end?

Trends and disparities in health at the end of life

PONE-D-21-16309R2

Dear Dr. Vierboom,

We’re pleased to inform you that your manuscript has been judged scientifically suitable for publication and will be formally accepted for publication once it meets all outstanding technical requirements.

Kind regards,

Ellen L. Idler

Academic Editor

PLOS ONE

Additional Editor Comments (optional):

One of the reviewers for this round is still requesting additional line-editing before final submission, for precision and more use of active voice. Please see the reviewer comments below.

Reviewers' comments:

Reviewer's Responses to Questions

**Comments to the Author**

1. If the authors have adequately addressed your comments raised in a previous round of review and you feel that this manuscript is now acceptable for publication, you may indicate that here to bypass the “Comments to the Author” section, enter your conflict of interest statement in the “Confidential to Editor” section, and submit your "Accept" recommendation.

Reviewer #1: All comments have been addressed

Reviewer #2: All comments have been addressed

2. Is the manuscript technically sound, and do the data support the conclusions?

Reviewer #1: Yes

Reviewer #2: Yes

3. Has the statistical analysis been performed appropriately and rigorously? 

Reviewer #1: Yes

Reviewer #2: Yes

4. Have the authors made all data underlying the findings in their manuscript fully available?

Reviewer #1: Yes

Reviewer #2: Yes

5. Is the manuscript presented in an intelligible fashion and written in standard English?

Reviewer #1: Yes

Reviewer #2: Yes

6. Review Comments to the Author

Reviewer #1: (No Response)

Reviewer #2: The authors have done a highly responsive revision, and the main methodological and conceptual concerns have been adequately addressed. I have lingering concerns about the writing, which could be more direct and precise in places. I would encourage the use of active v. passive voice where possible, and avoiding the use of vague phrases that add little to the text. For instance, rather than referring to "various populations" just specify precisely what the populations are. No need to use phrases like "in this paper" (as it is clear that the proposed analyses are being done in your paper).

I encourage a very careful line-edit to enhance the clarity and conciseness of the work. Congratulations on a successful revision.

7. PLOS authors have the option to publish the peer review history of their article (what does this mean?). If published, this will include your full peer review and any attached files.

Reviewer #1: No

Reviewer #2: No

---

## [Editor Report · Acceptance letter]

30 Jun 2022

PONE-D-21-16309R2 

How does it all end?
Trends and disparities in health at the end of life 

Dear Dr. Vierboom:

I'm pleased to inform you that your manuscript has been deemed suitable for publication in PLOS ONE. Congratulations! Your manuscript is now with our production department. 

Kind regards, 

on behalf of

Professor Ellen L. Idler 

Academic Editor

PLOS ONE